# Impact of Planting Date and Insecticide Application Methods on *Melanaphis sorghi* (Hemiptera: Aphididae) Infestation and Forage Type Sorghum Yield

**DOI:** 10.3390/insects13111038

**Published:** 2022-11-10

**Authors:** Osariyekemwen Uyi, Francis P. F. Reay-Jones, Xinzhi Ni, David Buntin, Alana Jacobson, Somashekhar Punnuri, Michael D. Toews

**Affiliations:** 1Department of Entomology, University of Georgia, 2360 Rainwater Rd., Tifton, GA 31793, USA; 2Department of Animal and Environmental Biology, University of Benin, PMB 1154, Benin City 300001, Nigeria; 3Department of Zoology and Entomology, Faculty of Natural and Agricultural Sciences, University of the Free State, P.O. Box 339, Bloemfontein 9300, South Africa; 4Department of Plant and Environmental Sciences, Clemson University, Pee Dee Research and Education Center, 2200 Pocket Rd, Florence, SC 29506, USA; 5USDA-ARS, Crop Genetics and Breeding Research Unit, 2747 Davis Road, Tifton, GA 31793, USA; 6Department of Entomology, University of Georgia, 1109 Experiment St., Griffin, GA 30223, USA; 7Department of Entomology and Plant Pathology, Auburn University, Auburn, AL 36849, USA; 8College of Agriculture, Family Sciences and Technology, Fort Valley State University, 1005 State University Dr., Fort Valley, GA 31030, USA

**Keywords:** insecticide efficacy, aphid infestation, foliar application, in-furrow application, insect-plant interactions, early planting, late planting

## Abstract

**Simple Summary:**

Infestation of forage sorghum by *Melanaphis sorghi* remains a major threat to silage production in the southeastern USA. Studies aiming to refine IPM strategies are crucial to improve management of this invasive pest. Here, we investigated the impact of planting date and insecticide application methods, including in-furrow vs. foliar applications, on *M. sorghi* infestation and grain sorghum yield in Tifton, GA, and Florence, SC, USA. In-furrow applications of *flupyradifurone* significantly suppressed aphid infestations in both the early and late planted sorghum at both study locations in both years, while aphid populations were also successfully suppressed following the foliar application of the insecticide after threshold numbers of aphids per leaf were reached. Early planting and in-furrow insecticide application improved yield the most at Florence in the 2020 study. However, in years of low aphid abundance, no difference in yield was observed between treatments, including untreated. This research demonstrates that early planting and in-furrow and foliar insecticide applications provided sufficient protection to prevent significant yield loss. Thus, this suggests that early planting of forage sorghum combined with in-furrow and foliar insecticide applications can suppress aphid infestations and improve silage production in southern USA.

**Abstract:**

Studies on the management of the invasive *Melanaphis sorghi* are essential to refining integrated pest management strategies against *M. sorghi* in forage sorghum in the USA. The objective of this study was to determine the impact of planting date (early planting and late planting) and in-furrow and foliar insecticide application of *flupyradifurone*, on *M. sorghi* infestation and forage sorghum yield in Tifton, Georgia and Florence, South Carolina, USA, in 2020 and 2021. Early planted sorghum supported slightly higher aphid density and severity of infestation as evident in the greater cumulative insect days values in the early planted sorghum at both Florence and Tifton in 2020 and 2021. A single foliar application reduced aphid infestations below the threshold level of 50 aphids per leaf. In contrast, in-furrow insecticidal application in selected plots at both locations significantly suppressed *M. sorghi* density to near-zero levels. Yield results in Florence in 2020 showed that sorghum yield was over 50% greater in early planted plots compared to late planted plots. Both insecticide treatments (foliar and in-furrow) resulted in significantly higher yield than untreated plots. These data indicate that early planting coupled with in-furrow and foliar insecticide applications can suppress *M. sorghi* infestations and improve silage production in forage sorghum in the USA.

## 1. Introduction

*Melanaphis sorghi* (Theobald) (Hemiptera: Aphididae) (previously known as *Melanaphis sacchari* Zehntner) is an invasive insect pest that poses a serious threat to sorghum production across the southern United States of America (USA) [1,2,3,4]. The multivoltine *M. sorghi* has emerged as a serious pest of grain, silage, and forage sorghum in the USA since 2013, when a new haplotype was first detected in Texas [1,5,6]. 

Despite the threat posed by *M. sorghi* to forage sorghum production in southern USA, there are surprisingly few published studies investigating the ecology and management of this invasive pest in forage sorghum fields (for exceptions, see [7,8,9]). In contrast, numerous empirical studies in grain sorghum fields across southern US states have documented the population dynamics of *M. sorghi* and yield under a range of cultivars, insecticide application regimes, and production practices [3,4,10,11]. For example, host plant resistance studies have led to the identification of commercial grain sorghum hybrids that are resistant to *M. sorghi* [4,12,13]. These studies revealed the rapid population growth of *M. sorghi* and documented up to 100% yield loss in unprotected grain sorghum [4]. Seiter et al. [10] investigated the effects of insecticide application and planting date on *M. sorghi* population dynamics and grain sorghum yield. The limited literature on the damage potential of *M. sorghi* in forage sorghum suggests that aphid feeding damage, honeydew production by aphids, and the resultant sooty mold growth on leaf surface may increase lodging and reduce forage quality, nutritive value, and yield in Texas, due to reduced photosynthesis [7]. In a recent study, Bell et al. [8] showed that increased aphid numbers resulted in higher plant damage and yield losses. To prevent yield and nutritive value losses in silage, the authors suggested that *M. sorghi* should be controlled when ≥20% of the leaf area is infested. 

The development of economic thresholds, identification of resistant grain sorghum hybrids and planting date manipulation as a cultural practice for grain sorghum production [4,10,13,14] provide a foundation for the development of integrated pest management strategies against *M. sorghi*. There is currently little published research on the effect of production practices such as choosing planting dates and applying insecticides to improve forage sorghum yield, despite the noticeable damage potential of *M. sorghi* in forage sorghum in southern USA. The foliar application of *flupyradifurone* will suppress *M. sorghi* populations in both grain and forage sorghum [4,8,15], but the efficacy of this insecticide and ability to apply over a tall forage crop varies with geographic locations and climatic conditions. Therefore, the objective of this research was to examine the effects of planting date and insecticide application (two different dosages of in-furrow (IF): low and high IF, and foliar application) on *M. sorghi* infestations and forage sorghum yields at two representative locations in the southeastern Coastal Plain Region of the USA.

## 2. Materials and Methods

### 2.1. Study Locations, Design, and Agronomic Practices

The impact of planting date and different methods of insecticide application (in-furrow and foliar insecticide applications) on *M. sorghi* densities and forage sorghum yield were evaluated over two growing seasons (2020 and 2021) at Tifton, GA (31.5120° N, 83.6434° W) and Florence, SC (34.3650° N, 80.0088° W), USA. In spring of 2020 and 2021, plots of a forage sorghum hybrid (SP1615, Sorghum Partners, S&W Seed Co, Longmont, CO, USA) were established at Tifton and Florence. A split-plot design was used for each experiment with planting date as the main plot factor and insecticide as the subplot factor. After spreading the recommended amounts of dry fertilizer, the fertilizer was incorporated using a field cultivator, and then, seedbed preparation was accomplished with a one-pass ripper bedder with the subsoil shank set to a depth of 50.8 cm for breaking the hardpan under the rows. All plots were delineated and planted using a vacuum planter at a density of 186,186 plants per ha. In Tifton, the planting dates were 17 April and 28 May in 2020 and 7 May and 11 June in 2021 trials; plots measured four rows of 12.2 m (0.91-m row spacing) in Tifton. In Florence, the planting dates were 11 May and 10 June in 2020 and 20 May and 16 June in 2021; plots measured 4 rows of 15.2 m (0.76 m row spacing). The four insecticide treatments were untreated, 117 g/ha flupyradifurone in-furrow (IF), 146 g/ha flupyradifurone IF, and 73 g/ha flupyradifurone + adjuvant as a foliar application. Sivanto HL (Bayer CropScience LP, St. Louis, MO, USA) was used for the in-furrow and foliar applications in 2020; however, that formulation was cancelled by the registrant for the 2021 crop year, so we used Sivanto Prime instead. Regardless of the formulation, the AI and rates were identical across the years. 

### 2.2. Insecticide Treatment and Aphid Infestation Assessment

The in-furrow insecticide treatments were applied at planting in selected plots at both locations. Microjet applicators were used to apply a total of 57.5 L of solution per ha directly on the seed in front of the disk closer. At both locations, designated foliar plots received a one-time application of flupyradifurone at 73 g/ha, administered using a self-propelled sprayer equipped with hollow cone nozzles (model TXVS-8, TeeJet Technologies, Spraying Systems Co., Glendale Heights, IL, USA). Applications were delivered in a spray volume of 93.5 L/ha. *Melanaphis sorghi* infestations were estimated by enumerating aphids, regardless of age, on each sampled leaf. In Tifton, sugarcane aphids were counted on six lower and 6 upper leaves of 6 randomly selected sorghum plants per plot each week. In Florence, sugarcane aphids were counted on 10 lower and 10 upper leaves of 10 randomly selected sorghum plants per plot each week. Assessments started four weeks after planting and continued until the grain reached the hard dough stage.

The timing of the foliar flupyradifurone application was determined using an economic threshold of 50 aphids per lower leaf, which had previously been developed for grain sorghum [4]. When the aphid population across the unsprayed plots reached 50 aphids per bottom leaf, a rescue foliar insecticide treatment was initiated in designated plots. To determine whether plots reached threshold, the mean number of *M. sorghi* on bottom leaves were first averaged across the 6 (Tifton) or 10 subsamples (Florence) per plot. To simultaneously account for aphid abundance and duration of infestations, aphid counts from top and bottom leaves were converted to cumulative insect days (CID) on a per plot basis following the methods of Ruppel [16]. Briefly, aphid days were calculated for each sampling interval as the mean density of two consecutive sample dates multiplied by the length of the interval between the dates in days. These values accumulated over the entire sampling period in each year, providing a cumulative estimate of aphid infestation intensity for each plot.

### 2.3. Harvest

In Tifton, the middle two rows of each plot in both the early planted and late planted sorghum were harvested for yield using a self-propelled forage chopper on 10 September 2021. In 2020, harvest was could not be collected due to equipment breakdown and lodging caused by a severe storm. In Florence, a 10-ft section of a middle row of each plot was hand harvested for yield on 13 October for both late and early planting dates in 2020. Because of an incorrect setting on the planter in the 2021 study, the stand count in the first planting date was poor; thus, yield data were not collected for the early planting date. In the late planted plots, a 10-ft section of a middle row of each plot was harvested for yield on October 12. Forage samples were weighed by plot and a 250 g subsample was dried at 80 °C for 48 h to estimate moisture content. Final yield estimates were extrapolated to kg/ha on a dry weight basis.

### 2.4. Data Analysis

The effect of planting date and insecticide applications on CID was log transformed and analyzed using a Generalized Linear Model (GLM) (assuming normal distribution with an identity link function). Our model included CID as dependent variable with insecticide treatment, planting date, and the interaction as fixed effects, and block and block × main plot as random effects. When the overall results were significant, differences among means were compared using the sequential Bonferroni test. The effects of planting date and insecticide treatments on forage sorghum yield were evaluated using univariate General Linear Model analysis of variance (GLM ANOVA). When an F-statistic was significant, the differences in means were compared using the Tukey’s Honest Significant Difference (HSD) test. All analyses were performed using IBM SPSS Statistical software version 20.0 (SPSS, Chicago, IL, USA).

## 3. Results

### 3.1. Aphid Population Trends

In 2020, aphids first appeared on 2 June (46 days after planting) in the early planted forage sorghum plots and on 23 June (26 days after planting) in the late planted plots in Tifton (Figure 1A,B). Aphid populations tended to be higher in the early planted plots compared to the late planted plots. Populations in the early planted plots exceeded the treatment threshold in the untreated plots within two weeks of first detection compared to plots that received in-furrow applications. In late planted plots, aphids exceeded the treatment threshold in the untreated plots within one week of detection. Foliar applications reduced the infestation below the threshold (Figure 1A,B). Untreated plots in both planting dates continued experiencing extremely high infestations (300+ aphids per leaf) for three weeks.

Aphid densities were generally lower in 2021 compared to 2020. Aphids first appeared on 15 June (38 days after planting) in the early planted plots and 23 June (12 days after planting) in the late planted plots in 2021 (Figure 1C,D). Aphid populations in the early planted plots exceeded the treatment threshold in the untreated plots within three weeks of first detection compared to plots that received in-furrow applications. Foliar applications made to selected plots in mid-July (early planting date) reduced the infestation below threshold (Figure 1C,D). In late planted plots, aphids were first detected in late June, but the population crashed to a near zero by 20 July without insecticide application (Figure 1D). Aphid populations never reached the treatment threshold in the high and low in-furrow applications in both the early and late planted plots in Tifton in 2021.

In 2020 in Florence, aphids first appeared on 25 June (33 days after planting) in the early planted plots and on 7 July (27 days after planting) in the late planted plots (Figure 2A,B). Aphid populations in the early planted plots exceeded the treatment threshold within two weeks of first detection. In late planted plots, aphids exceeded the treatment threshold in the untreated plots within one week following their first detection. Foliar application to designated plots reduced the infestation below threshold in both the early and late planted trials (Figure 2A,B).

In 2021, aphids first appeared on 24 June (35 days after planting) in the early planted plots and on 1 July (15 days after planting) in the late planted plots (Figure 2C,D). Aphid populations in the early planted plots exceeded the treatment threshold within two weeks of first detection. In the late planted plots, aphids exceeded the treatment threshold in the untreated plots within two weeks following their first detection. Foliar application to designated plots reduced the infestation below the threshold levels in both the early and late planted sorghum trials in both 2020 and 2021 (Figure 2C,D). Aphid populations never reached the treatment threshold in the high and low rates of in-furrow insecticide applications in both the early and late planted plots in Florence. The early planted plots recorded a higher number of aphids compared to the late planted plots.

### 3.2. Cumulative Insect Days

In Tifton, *M. sorghi* infestation severity as indicated by CID was significantly influenced by planting date and foliar insecticide application (Table 1). Generally, *M. sorghi* infestation was higher in early planted sorghum plots (compared to late planted plots) in 2020 and 2021 (Figure 3A,B). Aphid CID was highest in untreated early and late planted plots in 2020 and 2021 (Figure 3A,B). In 2020, CID values were more than 13-fold and 4-fold higher in untreated plots relative to both in-furrow insecticide treatments in early planted sorghum and late planted sorghum, respectively (Figure 3A). However, in the 2021 study, CID values were 18-fold and 5-fold higher in untreated plots compared to both in-furrow insecticide treatments in early planted sorghum and late planted sorghum, respectively (Figure 3B). In both 2020 and 2021, in-furrow insecticide application supported the fewest aphids. There was no significant difference in CID values between the high and low in-furrow treatments. In Florence, planting date and insecticide application significantly influenced aphid infestation in the 2020 and 2021 trials (Table 1). Aphid infestation was higher in early planted forage sorghum plots compared to late planted plots in both 2020 and 2021 (Table 1; Figure 4A,B). Aphid infestation (CID) was highest in untreated early and late planted plots in 2020 and 2021 (Figure 4A,B). In 2020, CID values were more than 222-fold and 368-fold higher in untreated plots relative to both in-furrow insecticide treatments in early planted sorghum and late planted sorghum, respectively (Figure 4A). However, in 2021, CID values were only 5-fold and 27-fold higher in untreated plots relative to both in-furrow insecticide treatments in early planted sorghum and late planted sorghum, respectively (Figure 4B). Generally, in-furrow insecticide application supported the least aphid infestation levels for both the 2020 and 2021 study at both locations.

### 3.3. Forage Sorghum Yield

In Tifton, in 2021, yield was unaffected by planting date (*F* = 2.34; df = 1,29; *p* = 0.1410) and insecticide application (*F* = 0.5241; df = 3,29 *p* = 0.6891) and there was no significant interaction between planting date and insecticide treatment (*F* = 0.25; df = 1,29; *p* = 0.8591) (Figure 5). In 2020 in Florence, planting date (*F* = 17.08; df = 1,31; *p* = 0.0015) and insecticide treatment (*F* = 3.31; df = 1,31; *p* = 0.0371) significantly influenced forage yield (Figure 6A); however, there was no significant interaction between planting date and insecticide treatments (*F* = 1.54; df = 1,31; *p* = 0.2314). Sorghum yield was over 50% greater in early planted plots compared to late planted plots. Compared to the insecticide treated plots, untreated plots recorded more than 20% and 50% yield loss in the early planted sorghum and late planted sorghum, respectively (Figure 6A). In 2021, yield data were unavailable for early planted forage sorghum plots. Insecticide application did not influence forage yield in the late planted plot (*F* = 0.61; df = 1,15; *p* = 0.5941) in 2021 (Figure 6B).

## 4. Discussion

The results of this study clearly indicate that in-furrow and foliar applications of flupyradifurone have the potential to effectively suppress aphid infestations and prevent injury due to aphid attack. This study demonstrated for the first time, the benefits of combining early planting with in-furrow and foliar applications of *flupyradifurone* to control *M. sorghi* infestations in forage sorghum. A consistent pattern in aphid infestation levels was observed, except in 2021, where aphid numbers were generally low. However, early planting combined with in-furrow and foliar insecticide applications improved yield the most at Florence in the 2020 study. This study is important because it shows the high degree of efficacy of in-furrow and foliar insecticide application in suppressing aphid infestations. Our finding is vital for refining strategies for the integrated management of *M. sorghi* in southern USA.

The peak in *M. sorghi* populations that occurred between three and four weeks from the first date of infestation in both plant dates is similar to the findings of earlier studies on grain sorghum [3,10,14], where aphid populations exhibited the same phenomenon. However, the effect of planting date on aphid population densities and infestations was inconsistent; population densities and CID values were greater in early than in late planted plots in Tifton in 2020 and 2021, while reduced infestations were recorded in Florence in early planted sorghum compared to the late planted plots in 2020. Although the greater aphid population densities and CID values in late planted sorghum plots in 2020 in Florence contradict our findings in Tifton, it supported the findings of Seiter et al. [10] and Szczepaniec [3], who found lower aphid densities and infestations in early planted sorghum compared to sorghum with late or conventional planting dates in Texas and Arkansas. While the high aphid numbers and infestations in the early planted sorghum at Tifton may seems incongruous, it should be noted that aphid population and the severity of plant injury due to aphid infestations and exponential aphid population increase may likely be influenced by latitude, seasonality, and rainfall. For example, Lahiri et al. [4] reported variations in *M. sorghi* infestation intensity on grain sorghum among locations and years in southeastern USA. The generally low population densities (less than 100 aphids per leaf in both early and late planted sorghum) and low CID values of *M. sorghi* in the 2021 study in Tifton may be due to the high unusual rainfall.

Compared to untreated, foliar insecticide application of low and high in-furrow rate of *flupyradifurone* significantly suppressed aphid infestation in both the early and late planted sorghum at both study locations in both years. Thus, in-furrow and foliar *flupyradifurone* applications are efficient chemical control methods of managing *M. sorghi* infestations in forage sorghum. The generally consistent low CID values in in-furrow treated plots in Tifton and Florence over the two years demonstrate the reliability and efficacy of this method of insecticide application in the management of *M. sorghi* across a wide geographic range in the USA. While the use of foliar insecticide to suppress *M. sorghi* populations have been previously demonstrated in forage sorghum and other types of high biomass sorghum production, e.g., sweet and bioenergy sorghum [4,8], this is the first report demonstrating the effectiveness of in-furrow insecticide application in the management *M. sorghi* in forage sorghum. In-furrow applications are also simpler because timing is not an issue and there are no issues as with foliar application by ground equipment over an impressively tall (3–3.7 m) standing crop. 

In-furrow application of insecticides have been employed in suppression of pests including invasive aphids [17,18]. For example, Howell and Reed [17] evaluated the efficacy of in-furrow insecticide application for thrips and aphid (*Aphis gossypii* Glover) control in cotton and found that aphid and thrips control was excellent in all four different in-furrow treatments. The efficacy of the in-furrow application in our study did not extinguish the effectiveness of foliar insecticide application, as foliar applications of *flupyradifurone* were also effective at reducing *M. sorghi* population to near zero levels in the early and late planted sorghums in both Tifton and Florence, which is consistent with what has been reported for grain sorghum [4,14,19]. Because aphid populations collapsed after the foliar applications of *flupyradifurone* in both the treated and untreated plots in 2021 in Tifton, the foliar application alone may not be responsible for the decline in aphid populations. Sudden *M. sorghi* populations declines, and even collapse, have also been observed in other sorghum studies throughout the southeastern US states [4,11] 

The reduced aphid infestations in the early planted sorghum in 2020 in Florence significantly increased forage sorghum yield; late planted sorghum suffered over 50% yield loss. Although little is known about the effect of planting dates on forage sorghum yield, a recent study on grain sorghum in Dasha County, Arkansas, found that early planting significantly preserved grain yield because of reduced aphid pressure [10]. The applications of all three insecticide treatments (low and high in-furrow rates and foliar applications) in 2020 in Florence resulted in significantly higher forage sorghum yield than untreated plots, where yield was reduced by 35%. The lack of significant difference in yield between low and high in-furrow rates of insecticide suggest that growers may save money by applying less insecticide. The positive impact of insecticide applications was more evident in the early planted plots. This implies that a combination of early planting to reduce pest pressure and avoid unfavorable late summer heat and drought [20], and insecticide applications in forage sorghum may be key to preserving yield by growers in southeastern USA. While many studies have reported foliar insecticide application to preserve yield in grain sorghum in Arkansas, Georgia, Texas, South Carolina, and Louisiana [3,4,10,11], only a few have actually demonstrated this in forage sorghum (e.g., [8]). To our knowledge, this is the first study that attempts to demonstrate the effect of in-furrow insecticide application to manage *M. sorghi* in forage sorghum. 

This research demonstrates that early planting and in-furrow and foliar insecticide applications provided sufficient protection to prevent significant yield loss. Thus, this suggests that early planting of forage sorghum and in-furrow insecticide application is the most consistent way to suppress aphid infestations and improve silage production in southern USA. Within the context of IPM, proactive applications of pesticides, such as the use of in-furrow application, are only recommended if pest pressure is predictably high enough to justify their use [21]. While a combination of in-furrow insecticidal application may be of benefit to growers, growers should consider manipulating sorghum hybrids and planting date when making decisions on the use of in-furrow applications of *flupyradifurone*. Historically, corn has been the main silage crop in the USA, although recent reports suggest that forage sorghum has the potential to produce large quantities of silage that can sustain dairy and feedlot beef cattle operations [22,23]. Therefore, more studies that aim to optimize biomass production of other types (e.g., silage, forage, sweet, and bioenergy) of sorghum by effectively suppressing *M. sorghi* infestations are needed across a wider geographical area. Such comprehensive understanding of *M. sorghi* population dynamics on all types of sorghum would allow us to further improve the integrated management tactics for the management of this invasive insect pest. 

## Figures and Tables

**Figure 1 insects-13-01038-f001:**
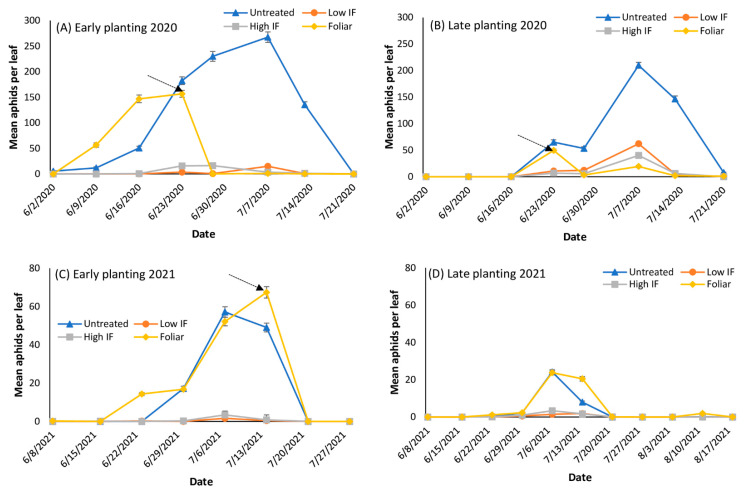
Mean number (±SE) of *Melanaphis sorghi* on bottom leaves for early (**A**) and late (**B**) planting in 2020 and for early (**C**) and late (**D**) planting in 2021 on grain sorghum in Tifton, GA. The arrow indicates timing of foliar insecticide application. Forage sorghum in the 2021 late planted sorghum did not receive a foliar insecticide application because *M. sorghi* populations never reached the threshold. The four insecticide treatments are: the untreated, low in-furrow (IF), high IF, and foliar applications with flupyradifurone.

**Figure 2 insects-13-01038-f002:**
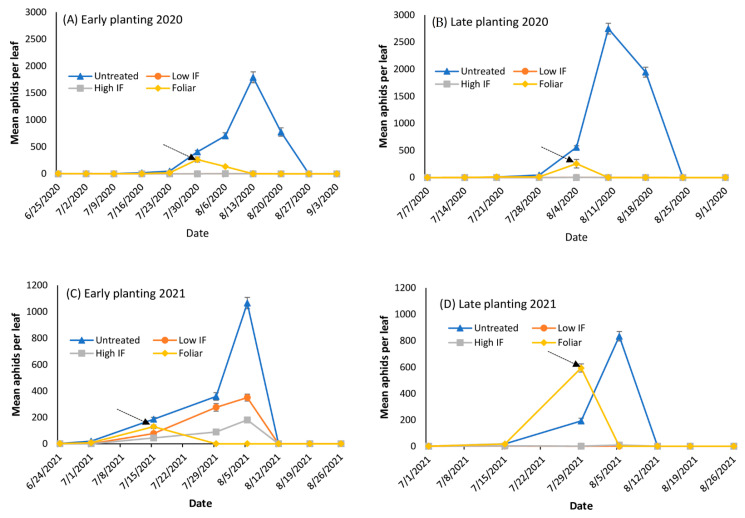
Mean number of *Melanaphis sorghi* on bottom leaves for early (**A**) and late (**B**) planting in 2020 and for early (**C**) and late (**D**) planting in 2021 on forage sorghum in Florence, SC. The arrow indicates timing of foliar insecticide application. The four insecticide treatments are: the untreated, low in-furrow (IF), high IF, and foliar applications with flupyradifurone.

**Figure 3 insects-13-01038-f003:**
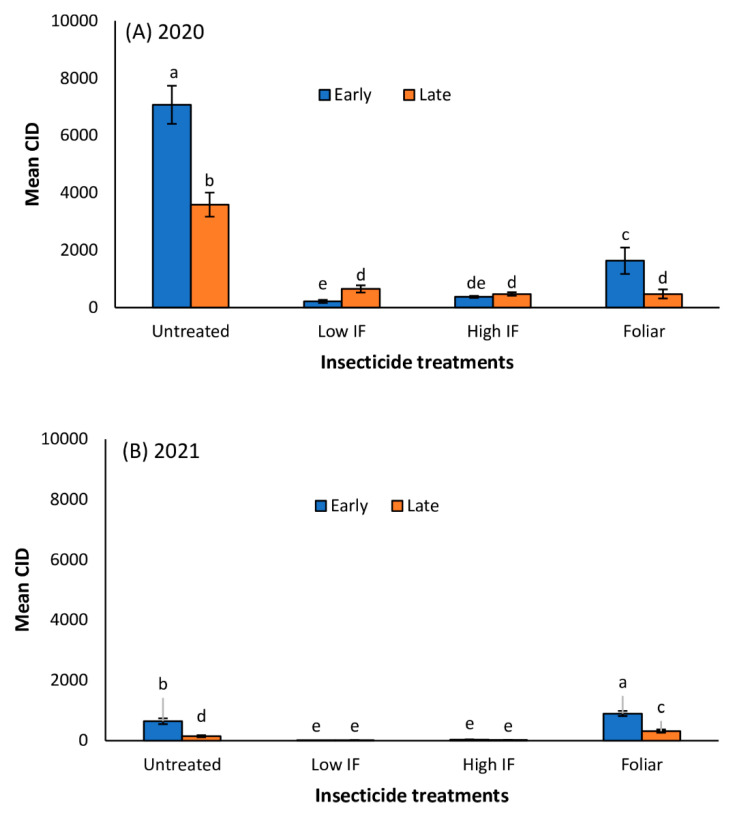
Mean (±SE) cumulative insect days (CID) in early and late planted forage sorghum treated with *flupyradifurone* in-furrow (IF) or foliar in 2020 (**A**) and 2021 (**B**) in Tifton, GA. Means with different letters are significantly different (sequential Bonferroni test, *p* < 0.05) among all four insecticide treatments and planting date.

**Figure 4 insects-13-01038-f004:**
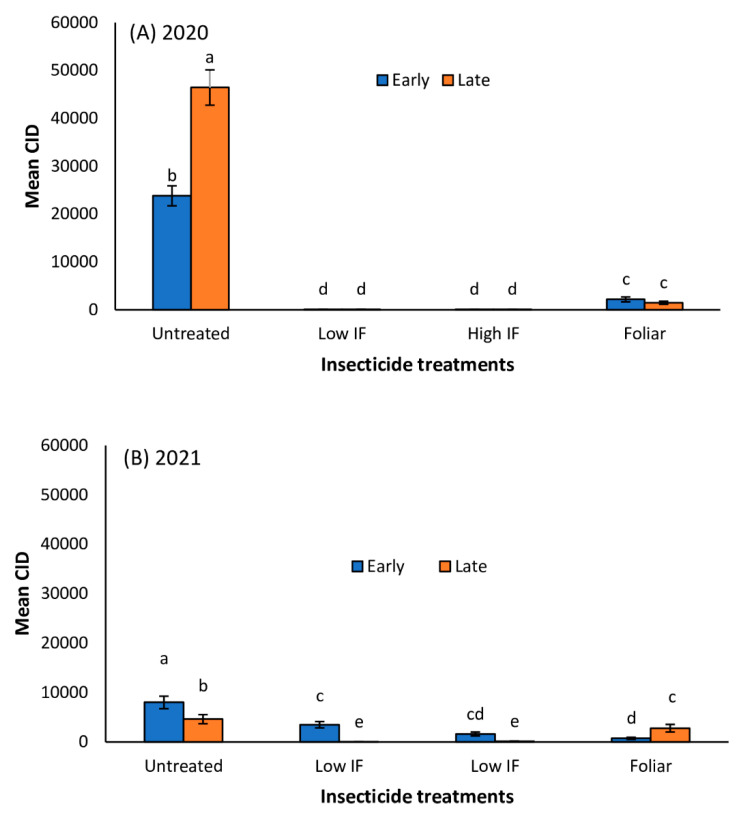
Mean (±SE) cumulative insect days (CID) in early and late planted forage sorghum treated with *flupyradifurone* in-furrow (IF) or foliar in 2020 (**A**) and 2021 (**B**) in Florence, SC. Means with different letters are significantly different (sequential Bonferroni test, *p* < 0.05) among all four insecticide treatments and planting date.

**Figure 5 insects-13-01038-f005:**
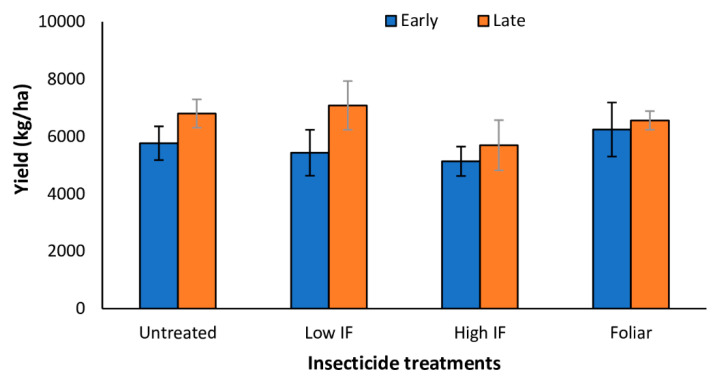
Mean (±SE) forage yield (kg/ha on a dry weight basis) in early and late planted forage sorghum treated with *flupyradifurone* in-furrow (IF) or foliar in 2021 in Tifton, GA. There were no differences in yield between both planting dates and among insecticide treatments (*p* > 0.05; Turkey’s HSD test). Note: forage sorghum in the 2020 study were not harvested due to lodging and equipment breakdown.

**Figure 6 insects-13-01038-f006:**
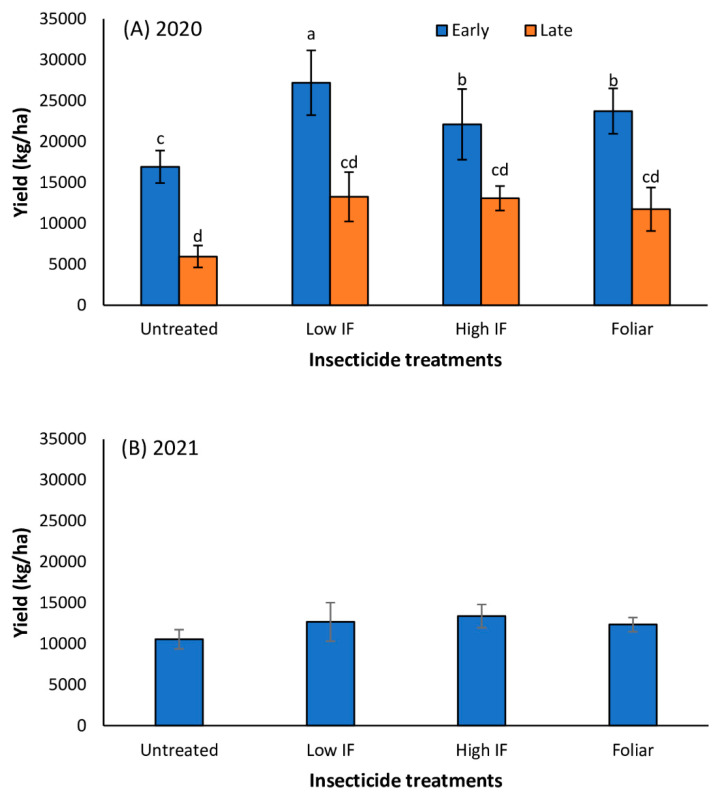
Mean (±SE) forage yield (kg/ha on a dry weight basis) in early and late planted forage sorghum treated with *flupyradifurone* in-furrow (IF) or foliar in 2020 (**A**) and 2021 (**B**) in Florence, SC. Bars with different letters are significantly different (*p* < 0.05; Turkey’s HSD test) among all four insecticide treatments and the two planting dates. Note: forage sorghum in early planted plots in the 2021 study were not harvested due to error at planting.

**Table 1 insects-13-01038-t001:** Generalized linear model (GLM) results for effects of planting date, insecticide applications and their interaction on cumulative insect days (CID) in the 2020 and 2021 study at Tifton, Georgia and Florence, South Carolina.

Location	Effect	d.f.	Wald χ^2^	*p*-Value
**Tifton**	2020 trial			
	Intercept	1	161.336	**0.0001**
	Planting date	1	12.981	**0.0003**
	Insecticide treatment	3	221.614	**0.0001**
	Planting date × insecticide treatment	3	31.491	**0.0001**
	2021 trial			
	Intercept	1	30.314	**0.0001**
	Planting date	1	8.093	**0.0041**
	Insecticide treatment	3	28.235	**0.0001**
	Planting date × insecticide treatment	3	6.101	0.5443

**Florence**	2020 trial			
	Intercept	1	342.988	**0.0001**
	Planting date	1	29.939	**0.0001**
	Insecticide treatment	3	886.861	**0.0001**
	Planting date × insecticide treatment	3	97.476	**0.0001**
	2021 trial			
	Intercept	1	88.507	**0.0001**
	Planting date	1	7.350	**0.0071**
	Insecticide treatment	3	54.830	**0.0001**
	Planting date × insecticide treatment	3	15.104	**0.0020**

The data analysis was conducted following log transformation; a normal distribution with an identity link function was assumed. Statistically significant values are indicated in bold.

## Data Availability

The data that are presented in this study are available in the article.

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
