# Peer review of "Impact of Planting Date and Insecticide Application Methods on Melanaphis sorghi (Hemiptera: Aphididae) Infestation and Forage Type Sorghum Yield"

_insects, 2022, doi:10.3390/insects13111038_

Round 1
Reviewer 1 Report
I found the study generally interesting and well presented and any comments can be found in the attached document. I have, however, some issues with the message presented in the summary & abstract (which is what most people will read) which advocate the use of in-furrow treatment as the best option. This study is based on only 2 farms and in 2 out of 3 yearly samples that could be evaluated, no significant difference in yield was found between the treatments and there was only a slight improvement in yield when comparing in-furrow vs foliar in the third one. Thus I am not convinced that there is a real indication it works better than foliar application of the crop once threshold numbers are reached and I feel it needs to be presented better as there are obvious drawbacks to continuous preventative application of pesticides.

Author Response
Reviewer #1:
I found the study generally interesting and well-presented, and any comments can be found in the attached document. I have, however, some issues with the message presented in the summary & abstract (which is what most people will read) which advocate the use of in-furrow treatment as the best option. This study is based on only 2 farms and in 2 out of 3 yearly samples that could be evaluated, no significant difference in yield was found between the treatments and there was only a slight improvement in yield when comparing in-furrow vs foliar in the third one. Thus, I am not convinced that there is a real indication it works better than foliar application of the crop once threshold numbers are reached and I feel it needs to be presented better as there are obvious drawbacks to continuous preventative application of pesticides.
We have largely improved our discussion on yield and stressed the fact that in-furrow insecticidal applications had a less consistent impact across locations and years. While yield seems like it should the ultimate response variable in crop production, it is highly subjective in small plot studies due to the fact that many stresses can affect yield during grain formation, ripening and drying. Small plot studies are more sensitive than whole fields to factors like late season extreme weather (wind and rainfall) and bird feeding leading up to harvest. All of these factors are more concentrated on field edges than in the center of a field. Further, black birds tend to concentrate and feed on the best plots (often earlier maturing) and rapidly confound yield estimates to the point that grain yield is not an effective response variable. Use of cumulative insect days as a response variable is the most precise way to estimate actual differences in insect pressure throughout the season.
Extreme variability in insect pressure across locations and between years was documented in this study using CID (range <100 to 7000 CID) as a response variable. These differences reflect real world differences in weather events that suppressed aphid populations as well as weather events like dry hot weather that exacerbated aphid populations. The ascertain that in-furrow applications are no better than foliar applications would be true if foliar applications could always be made on-time. However, in the real world, growers have to juggle supply chain availability of pesticides, mechanical breakdowns, weather events that prevent timely applications, and simultaneous treatment of fields in desperate locations. Each of these situations could result in the inability to make timely applications. Those concerns are ameliorated with the in-furrow application that always provided good protection without the risk of poor timing.
Below is our response to the comments and suggestions on the PDF attachment.
Attached PDF #1
Lines: 22-23: We have deleted lines 22-23 as suggested
Lines 23-26: Revised as suggested
Lines 26-30: Revised as suggested
Line 36: Corrected
Line 37: Corrected
Lines 42-44: Revised as suggested
Lines 90: Corrected
Line 95: Corrected
Lines 106-108: The planting dates of forage sorghum where chosen based on the agronomic advice from local extension agents. Typically, early planting ranges from mid-April to early May and late planting ranges from late May to mid-June. Planting operations was not uniform across years and locations because of severe weather conditions and heavy rainfall.
Line 112: Corrected
Lines 129-130: This was generally the case, but inclement weather prevented us from applying foliar insecticides in some occasions.
Line 172: Corrected
Line 174-175: Corrected
Line 216: Corrected
Line 239: Corrected
Line 246: Corrected
Line 280: no letters shown in Figure B because forage sorghum in early planted plots in the 2021 study were not harvested due to error at planting
Line 287-288:
289-291: Revised as suggested
Line 313: Corrected
Lines 315-326: Slightly revised to reflect the fact that in-furrow and foliar insecticide applications were effective in managing aphid infestations.
Lines 349-351:
Line 363: Corrected
Line 371: Corrected
Line 372: Corrected
Reviewer 2 Report
A good paper easy to follow and well written.
A few comments to help here:
Figure 1 and 2. Error bars are not visible
p value should be lower case and italicised; upper case P is statistical power not statistical significance.
Author Response
Reviewer #2:
A good paper easy to follow and well written.
Very many thanks for your positive comment
A few comments to help here:
Figure 1 and 2. Error bars are not visible
We have provided in the error bars in Figures 1 and 2
p value should be lower case and italicized; upper case P is statistical power not statistical significance.
Corrected
Very many thanks for your comments. We have now made all the requested minor corrections
Reviewer 3 Report
Title: Impact of planting date and insecticide application methods on 2 Melanaphis sorghi (Hemiptera: Aphididae) infestation and for- 3 age type sorghum yield
Authors: Osariyekemwen Uyi , Francis P. F. Reay-Jones , Xinzhi Ni , David Buntin , Alana Jacobson , Somashekhar Punnuri , and Michael D. Toews
General comments:
I think the scientific merits of manuscript titled: Impact of planting date and insecticide application methods on Melanaphis sorghi (Hemiptera: Aphididae) infestation and forage type sorghum yield are very good. The work is well structured and presented. I did not find any mistakes. Presented by authors data provide important information explaining the interaction of sorghum plants aphids. This kind of studies are very important for planning the management of the invasive insects as Melanaphis sorghi in the USA and across a wider geographical area and allow us to further improve the integrated management tactics for this invasive insect pest.
Therefore, the subject matter of the work is consistent with the profile of the Insects Journal.
I have a recommendation to standardize the font in Materials and Methods sections line 133. Please, adjust the citation in the text and the list to the requirements of the editorial office (“References must be numbered in order of appearance in the text (including table captions and figure legends) and listed individually at the end of the manuscript”).
Author Response
Reviewer #3:
I think the scientific merits of manuscript titled: Impact of planting date and insecticide application methods on Melanaphis sorghi (Hemiptera: Aphididae) infestation and forage type sorghum yield are very good. The work is well structured and presented. I did not find any mistakes. Presented by authors data provide important information explaining the interaction of sorghum plants aphids. This kind of studies are very important for planning the management of the invasive insects as Melanaphis sorghi in the USA and across a wider geographical area and allow us to further improve the integrated management tactics for this invasive insect pest.
Therefore, the subject matter of the work is consistent with the profile of the Insects Journal.
Very many thanks or your positive comments
I have a recommendation to standardize the font in Materials and Methods sections line 133. Please, adjust the citation in the text and the list to the requirements of the editorial office (“References must be numbered in order of appearance in the text (including table captions and figure legends) and listed individually at the end of the manuscript”).
Many thanks for the comments. We have now made the suggested changes to the citation and references.